# Organic amendment plus inoculum drivers: Who drives more P nutrition for wheat plant fitness in small duration soil experiment

Saba Ahmed[1], Nadeem Iqbal[2], Xiaoyan Tang[3], Rafiq Ahmad[1], Muhammad Irshad[1], Usman Irshad[1]*

1 Department of Environmental Sciences COMSATS, University Islamabad, Abbottabad, Pakistan, 2 Doctoral School of Environmental Sciences, University of Szeged, Szeged, Hungary, 3 College of Resources Sichuan Agriculture, University Chengdu, Chengdu, China

* usmanirshad@cuiatd.edu.pk

**Data Availability Statement:** The minimal data set is within the Supporting Information file S3_raw_data. The file includes the values behind the means, standard deviations and other

## Abstract

Functioning of ecosystems depends on the nutrient dynamics across trophic levels, largely mediated by microbial interactions in the soil food web. The present study investigated the use of phosphate solubilizing bacteria (PSB) and poultry manure (PM) for maintaining labile P in the soil for an extensive fertility enhancement and as a substitution of chemical fertilizers. Based on the different P solubilizing capabilities of Bacillus and Pseudomonas, a quadruple consortium of *Bacillus subtilis*, *Bacillus cereus*, *Bacillus thuringiensis* and *Pseudomonas fluorescens*, and their grazer nematodes (soil free living) supplemented with PM were studied. This study was carried out on the trophic levels of soil communities to assess the growth and availability of P to the wheat plants. Experiment was performed for 90 days. Comparing the unamended and amended predator results showed that nematode addition beyond bacterial treatment substantially increased the net available P by ≈2 times, and alkaline phosphatase (ALP) activity by 3.3 times. These results demonstrated the nematodes association with increasing nutrient availability or P mineralization. The interactive effect of PM as substrate and biological drivers was more noticeable on plant dry biomass (1.6 times) and plant P concentration (3.5times) compared to the similar unamended treatment. It is concluded that the biological drivers significantly enhanced the soil ALP and available P while the substrate and biological drivers enhanced dry biomass and plant P concentration. Bacterivore nematodes enhanced the effect of PSB for P mineralization via microbial loop and could be used for the enhancement of wheat production.

## Introduction

The function of soil microflora and fauna is very important in ecosystem services such as decomposition and nutrient dynamics across trophic levels which depends on their interactions in soil food web. These important biological drivers: bacteria, fungi and nematodes act as major sink and source of key abiotic components such as carbon, phosphorus and pH. Therefore, the deficiency of either component ultimately reduces output of soil ecosystem such as

measures reported and the values used to build graphs and statistical analysis. The accession numbers with URLs are as, a-MK418810 (https://www.ncbi.nlm.nih.gov/nuccore/MK418810.1/) b-MK417798 (https://www.ncbi.nlm.nih.gov/nuccore/MK417798) c-MK417800 (https://www.ncbi.nlm.nih.gov/nuccore/MK417800) d-MK418218 (https://www.ncbi.nlm.nih.gov/nuccore/MK418218).

**Funding:** The research project was supported by a grant from the Higher Education Commission of Pakistan via an NRPU grant No: 20-3655/R&D/HEC/14/400. The funders had no role in study design, data collection and analysis, decision to publish, or preparation of the manuscript.

**Competing interests:** I have read the journal's policy and the authors of this manuscript have no competing interests.

plant yield [1]. It has been reported that a limited availability of the phosphorus to plants significantly affected microbial diversity, plant growth and ecosystem prolificacy [2].

Phosphorus is one of the key abiotic regulators of ecosystem functions being indispensable for development and growth of plant. Agriculture soils are strongly depleted in available P for plants, although a sufficient 50–65% amount of total P is present but in complex organic and inorganic forms [3]. Also, substantial P turnover occurs in the surface layers of soil and microbial number reduce along soil depth. Various P management strategies were executed to maintain adequate labile P in the agricultural fields [4]. For that many ecosystems globally are experiencing mineral and organic phosphate fertilizer inputs begetting accumulation of stable P minerals and organic compounds in soil through biological assimilation and chemical processes of precipitation and sorption involved in P fixation. Consequently, depending on the soil nature much (80–90%) of this remains unavailable for plant uptake leaving approximately 10–20%/(2–10 μM) for plant use in the form of mono and diorthophosphate in soil solution [4]. Soil organic phosphorus is 30–70% of the total P, largely in the form of stable inositol phosphate (soil phytate), easily hydrolysable microbial biomass P account for (2–5) % [4]. The availability of these pools for plant use relies on the soil fertility [4,5].

Livestock manures, plant residues and compost are some organic amendments and have been proposed as alternatives or supplement to mineral fertilizers. These organic amendments effect microbial biomass and their diversity with consequent long term potential benefits of improved soil nutrient turnover and other essential ecological functions such as extracellular enzyme activities e.g., phospahatase enzyme production. However, these benefits of nutrient dynamics such as P availability for plant growth remains variable depending on the different farming systems [5–7]. Therefore, Inoculation of PSBs with or without organic (such as poultry manure) [6] or inorganic fertilizers [7] has been the research concern on P dynamics often with differing responses on crop yield and P availability [7,8]. Poultry manure is a concentrated potential source of P and contains more stable mineral-associated P compared to other manures [8–10]. Soil organisms respond in different ways to phosphorus availability due to different requirements of nutrients and economic strategies [5–11]. For instance, rhizosphere bacteria incorporate large amount of P in their biomass corresponding to high available C inputs in the form of manure. Thus reduced P availability is subject to competition for plant and microbial uptake depending on the residence time of P in their microbial biomass. Microbial assimilation of soil solution orthophosphate and its timely and expedient release according to the need of the plant or future generation of microorganisms lessens the soil P fixation by chemical sorption [11]. Depending on the soil and fertilizer input the release of P from microbial biomass can be as lowest as10 days to 170 days to one year maximum reported [3], which generally constitute a prospective source of available P for plants, especially in P deficient systems. Among various phosphorus management practices, strongly positive effects of PSBs (phosphate solubilizing bacteria) have widely been recognized as key drivers for a better plant growth and soil P availability. Phosphate solubilizing bacteria (PSBs) improve availability of P for plants [10,11]. Moreover, a synergistic and complementary microbial consortium based on different functional groups of the comprising strains may effectively enhance plant growth and effect of organic fertilizers such as poultry manure [12]. The mechanism to solubilize inorganic complexes is the production of organic acids and enzymes to hydrolyze the organic compounds (PMBs) phosphorus mineralizing bacteria [11,12]. The higher effect of Bacillus and Pseudomonas inoculation amongst other PSBs have been reported on wheat (*Triticum aestivum*), chickpea and other crops development and P availability [13–15]. The enhanced effect of labile P status in soil is by the production of hydrolyzing enzymes and organic acids following the process of mineralization and solubilization of organic compounds and mineral complexes of P, respectively [16].

Soil bacterivores nematodes are the important indicators of soil ecosystem functioning comprising 60–80% of total nematodes [17]. The presence of higher trophic level interaction such as bacterivores nematodes in the rhizosphere soil can directly affect the net P mineralization through excretion products or indirectly by feeding upon rhizosphere bacteria [17,18]. Therefore, bacterivores nematodes beneficially affect plant by releasing nutrients such as P from bacteria after feeding upon them [19]. This mechanism of interplay between microflora and fauna is known as, the 'microbial loop 'triggered by high rhizosphere microbial activity and biomass through carbon pulses provided as root exudates, transmit energy to following microfaunal grazers [17–19], where approximately 30-times increase in the numbers of free-living nematodes such as bacterial feeders may occur. During the growth of nematodes large amounts of available nutrients such as P, C and N are incorporated in their biomass on the other hand biomass production through grazing utilize 50–70% of the prey carbon [20].

The P mineralization potential by co-inoculation of single bacteria (Pseudomonas, Bacillus) +bacterivores nematodes was assessed on phytate source, detritus and organic matters [17–20]. But co-inoculation of bacterial consortia and nematodes have not been investigated in the presence of organic amendment in the wheat (*Triticum aestivum*) rhizosphere. These works demonstrated the co-inoculation of nematodes with single PSB strain or PSB single and consortium inoculation on P dynamics. The activity of microbial exoenzymes responsible for organic P mineralization can be strengthened through microbial grazers such as nematodes [18,19]. In a research study higher effect of bacteria and their grazer nematodes on rate of organic matter utilization on the basis of organic P mineralization was observed compared to bacteria alone effect [20].

There are many integrated soil fertility management strategies available, but more efforts are always needed to improve the various functions of PSBs and interaction with microfauna besides using organic wastes as nutrient resource. Therefore, the objective of the present study was to explore the complementary effects of PMBs and their grazer nematodes within multi-trophic interactions in the presence of poultry manure on soil available P, and wheat biomass.

## Material and method

### Soil sampling and study area

Soil for microbial isolation was collected from forest of Nathia Gali Abbottabad, lower Himalayan region in Khyber Pakhtunkhwa (KPK) Pakistan, Asia 34˚ 4' 20" North, 73˚ 23' 55" East, owing to the undisturbed soil ecosystem. For each rhizosphere soil sample, the soil stuck to the pine plant roots were collected to make composite sample, sealed and placed in an iced container right after sampling [21]. This composite soil sample was used to perform bacterial isolation, soil initial pH, water available phosphorus and total phosphorus.

### Isolation, screening and identification of PSBs

A synthetic solid medium Pikovskaya's (PVK) was prepared with the concentration (g)/L; Mg $SO_4$. $7H_2O$ 0.492 g, MOP's (pH stabilizer) 133.2 mL, Glucose 3.33 g, micro-nutrients 267 μL $KNO_3$ 0.100 g, calcium sulphate 0.543 g and calcium phosphate $Ca_3(PO4)_2$ 3.956 g were added as sole inorganic P source for being the most precipitated phosphorus mineral found in neutral and alkaline soils [17]. The organic P Pikovskaya's solid medium was added with (sodium phytate) $C_6H_6Na_{12}O_{24}P_6$ as sole organic P source instead of $Ca_3(PO_4)_2$ and MOPs for pH stabilization (pH 7–7.5) [18]. Following the dilution plate technique the isolation of PSBs was done. A sufficiently small 20μL aliquot from serially diluted ($10^{-4}$–$10^{-6}$) suspension of soil samples was spread on sterilized PVK solid medium and incubated for 48 hours at 30±2˚C. Purification was done by repeated single colony streaking on Pikovskaya's solid medium Then 110 strains

through the observation of halo zone around colonies on Pikovskaya's solid media qualitatively confirmed the inorganic phosphorus solubilization These isolates were further screened on the basis of organic phosphorus mineralization potential on Pikovskaya's solid medium with organic P sourc. e and 21 strains were confirmed as P (Po & Pi) mobilizers on the basis of halo zone formation [17,18].

## Biochemical assay for screening based on available inorganic P, pH and ALP activity indicators

The acquired 21 pure phosphorus mobilizing bacterial cultures were quantitatively evaluated for P (Po & Pi) mobilization on the basis of pH change, bacterial growth, orthophosphate concentration and alkaline phosphatase production "Table 1" [18,22,23]. For this all of strains were grown singly in Pikovskaya broth medium with sodium phytate as P source (pH 7.2) and sterile media without inoculation was set as blank. Bacterial growth was observed visually following incubation at 28˚C for 3 days in an incubator shaker (120 rpm) [18,19]. The bacterial cultures were kept 15–20 minutes to settle down insoluble culture media. Briefly, filtered (0.45 μm size) supernatant was taken, centrifuged 10000/min for 10 minutes at room temperature and diluted accordingly to determine available P and pH determination. pH was measured with pH meter while reagent 1 and 2 [24] was added following malachite green method for determination of available phosphorus. Standards were prepared to make calibration curve and inorganic P mineralized by bacteria from organic source were analyzed by microplate reader at 620 nm [24].

Alkaline phosphatase (ALP) activity was determined by taking 0.25 mL supernatant from fresh bacterial cultures; followed the ALP assay described for soil in the next section [22].

Based on these defining parameters of P mobilization i.e. pH, ALP activity and P mineralization 4 efficient strains out of 21 were selected for further analyses "Table 2" and sustained on solid organic Pikovskaya's medium.

## Molecular characterization of selected bacterial strains

The selected bacterial strains were identified using 16sRNA gene sequencing and submitted in gene bank with accession number "Table 2". The bacterial strains were stored in glycerol stocks at −80˚C and revived as and when required.

Presence of phosphorus mineralizing gene was done using q-PCR: In bacteria phoN, phnX gene encoding enzymes are responsible for organic P mineralization and gcd for inorganic P solubilization i.e., Genes encoding phoN included in bacteria responsible for enzymatic hydrolysis of organic P to orthophosphate during phosphate starvation [25]. The presence of responsible genes was analyzed in selected bacterial strains.

**Table 1. Listed is the selected efficient identified phosphorus mobilizing bacteria and their consortia with accession numbers, ALP activity and phosphorus mineralizing values.**

| Name Codes of bacteria | Names of bacterial strains | Accession Numbers | ALP Enzyme activity Moles of pNPP/ml/hr | Available P mg/ml |
|---|---|---|---|---|
| Bc | *Bacillus cereus* | MK418810 | 1.39 | 0.92 |
| Bs | *Bacillus subtilis* | MK417798 | 1.41 | 0.98 |
| Bt | *Bacillus thuringiensis* | MK417800 | 1.40 | 1.57 |
| Pf | *Pseudomonas fluorescens* | MK418218 | 1.42 | 1.40 |
| Bc+Bs+Bt+Pf | *Bacillus cereus+Bacillus subtilis+Bacillus thuringiensis+Pseudomonas fluorescens* | —— | 1.44 | 1.76 |

**Table 2. Showing the primers of the selected genes.**

| List of target specific primers, their amplicon length and nucleotide sequences used for gene presence identification. Enzyme | Amplicon length (bp) | Primer sequence 5'-3' | Primer name | Reference |
|---|---|---|---|---|
| Acid phosphatase (class A) | 159 | GGAAGAACGGCTCCTACCCIWSNGGNCA | phoN-FW | |
| | | CACGTCGGACTGCCAGTGIDMIYYRCA | phoN-RW | |
| Phosphonoacetaldehyde hydrolase | 147 | FWCGTGATCTTCGACtGGGCNGGNAC | phnX-FW | Berkgemper et al., 2016 |
| | | GTGGTCCCACTTCCCCADICCCATNGG | phnX-RW | |
| Quinoprotein glucose dehydrogenase | 330 | CGGCGTCATCCGGGSITIYRAYRT | gcd-FW | |
| | | GGGCATGTCCATGTCCCAIADRTCRTG | gcd-RW | |

## DNA extraction

DNA of each selected bacterial strain was extracted by adapted method from Irshad et al [18]. Supernatant from fresh cultures of selected bacterial strains was centrifuged and pellette was suspended in 500μL of 1X PBS repeatedly to remove all the media. after repeated thawing, tubes were centrifuged and the supernatant containing DNA fragments was treated with iso-propanol and washed with 70% ethanol and pellet was dried and dissolved in 1x phosphate buffer saline and kept at -80˚C [18].

PCR was performed to amplify the genes involved in P mineralization from each DNA extracts of selected PMB (phosphorus mobilizing bacteria); Bc, Bs, Bt and Pf, using primers of selected genes "Table 2". A ready to use master mix 10μl (containing Taq DNA polymerase, dNT P, MgCl$_2$, PCR buffer and PCR stabilizer), 1.5μL DNA extract of each strain, 1μL (R and F) primers, diluted to 20 μL final volume with molecular grade water H$_2$O. The reaction mixture was placed in thermocycler at condition according to Bergkemper et al. [25] Initial temperature (95˚C; 5F min) following denaturation; 30 cycles (95˚C; 1 min) then (57.6˚C; 1 min) annealing and elongation (72˚C; 30s) followed by a final step of elongation (72˚C; 4 min). This optimum annealing temperature of the primers was investigated in a preliminary test performing a gradient PCR (annealing temperature: phnX 57.6˚C, phoN 59˚C, gcd 55˚C [25]. Finally, PCR amplified product was evaluated by electrophoresis on 1.5% agarose gel in 1xTAE running buffer with 50bp to 1kb DNA ladder [25].

## Isolation of nematodes

Extraction of nematodes was carried out according to the Cobbs' sieving and decanting method as described by Irshad et al [26]. Following sequential sieving (sieves mesh size ranging from 20–325) and decanting to isolate soil free-living nematodes, the final backwashed material was spread on filter, supported on wire mesh and placed on water filled pertiplates at 22–25˚C for one week. Then nematodes move through filter in petriplate. Finally, the recovered extracted nematodes were counted and identified to bacterivores in 20ml of subsamples from pertiplate under microscope at magnification of 100· and 400· respectively. While a small aliquot was poured on sterilized TSA, incubated at 30˚C and maintained for further use.

## Microbial compatibility experiment

**Bacteria-bacteria compatibility.** The synergistic activity between the selected efficient isolates was performed following Cross Streak method [18–26]. According to this technique,

bacterial strains were cross streaked one following the other, then incubated at 30±2˚C 48 h. The cultured strains merged together with no reduced growth were considered compatible, while non-compatibility between them was assessed by growth suppression. A group of 4; *Bacillus cereus S₁₂ Bacillus subtillis*, *Bacillus thuringiensi*, *and Pseudomonas fluorescens* strains showed the higher compatibility for no growth suppression together. This shows their potential to be mixed as consortium to make bio inoculants.

**Bacterial nematode interaction** was assessed by the adapted method Hayat et al [27]. Briefly TSA sterilized large glass pertri plates were spot inoculated in center with the PSB isolates at 1 cm distance from each other, incubated overnight at 28±2˚C. Thereafter, 50 nematodes were emplaced towards the periphery at 2cm distance (approx.) from center. Two controls were also set with only bacteria at center and only nematodes with sterile water inoculation. The observation was made under microscope with nematodes movement path and bacterial colonies disturbance indicative of positive interaction between 4 selected isolates and nematodes.

## Preparation of inoculum

**Bacterial inoculum.**   Individual liquid cultures of each of four isolates; *Bacillus subtillis*, *Bacillus cereus*, *Bacillus thuringiensis* and *Pseudomonas fluorescens* were prepared in TSB media according to the adapted method [16], incubated overnight at 30±2˚C. These liquid cultures were repeatedly centrifuged (3000 rpm for 1 minute at room temperature) and suspended in sterile distilled water until thoroughly washed [18]. By taking small aliquots from these suspensions' cells were counted under microscope using cell counter and diluted accordingly. (Bc; 4.15x106 Bs; 3.9x107 Bt; 3.7x106 Pf; 4.45x105)/ml. Finally adjusted 1m of each of these suspensions (approximately equal no. of cells of each strain) were mixed to form inoculum pH (7.2) 4ml diluted to final volume 250ml.

**Inoculation.**   Bacterial consortia inoculation was done after 12 days of germination (15[th] day) of experiment with drop and pour method. A pierce was done in soil near the vicinity of roots and inoculated with 5 mL of bacterial inoculum with concentration ($5.25 \times 10^7$ cells/mL). Afterwards all the pots including uninoculated ones were watered with 10 mL of distilled water.

**Nematode inoculation.**   Nematodes were inoculated on 30[th] day of experiment. TSA Plate cultures of nematodes were washed with sterile distilled water and collected in falcon tube. Centrifuged and washed 3 times and resuspended in 5ml distilled water. Diluted and enumerated under microscope and 2ml of inoculant (1200 Ind/ml) were added in pots with drop and pour method after which all pots including uninoculated ones were watered with 10ml of distilled water [17].

## Experimental design

**Sampling and processing of soil and poultry manure.**   Microcosms were set up by first sampling, homogenizing, and processing the soil and poultry manure. The soil for the pot experiment was recovered from depth of 0 to 25 cm from a small crop field of COMSATS university Islamabad, Abbottabad campus (N 34˚ 13'22" and E 73˚ 16' 0"). Field was previously sown with maize and wheat on rotational basis. The soil contains loamy clay texture with deficiency in available plant P possessing slight alkaline properties.

The soil and fresh poultry manure samples were dried and sieved at 2 mm/10 mesh size [9]. Soil and processed poultry manure after initial analyses of total P, water available P and pH in triplicates "Table 3" were homogenized. According to the literature different rates of same amendments are applied from 1% to 10% on different basis. So the lower rate was considered

Table 3. Soil and poultry manure initial pH and available P analysis.

| Sample | pH | Water available P (mg/g) | Total P (mg/g) | Unavailable P (mg/g) |
|---|---|---|---|---|
| Soil | 7.79 | 0.314±0.2 | 98.81±5.61 | 98.496 |
| Poultry manure | 8.1 | 17.8±13.3 | 166.68±2.25 | 148.88 |
| Soil+Poultry manure | 7.86 | 60.41±1.2 | 269.29±4.21 | 208.88 |

for being economical [8–28]. Poultry manure (PM) as amendments was mixed with soil at 5% by weight i.e. 50g/kg.

Overall, 24 experimental units/pots were arranged with 2 biological treatments, 2 biological controls (four treatments), with and without poultry manure (PM) addition and triplicated each. To exploit trophic level, it was either added with bacteria/nematodes with and without wheat plant or added bacteria alone and with nematodes in wheat rhizosphere. We amended half of the microcosms with poultry manure while other half of the microcosms were not amended with poultry manure. Each small plastic pot was labelled and filled with 200 grams of soil/soil+PM mixture according to the treatments. Together this experimental setup created a high and low phosphorus resource environment in soil and allowed us to test trophic level with or without amendment on enzyme activity, and plant growth.

Afterwards, microcosms were placed in growth chamber in completely randomized design and maintained humidity (65%) and temperature (15–18°C) for optimal microbial activity [29] through watering 1-2ml distilled water with 16 hours day light and 8 hours night period, respectively.

## Pre and post analyses

**Phosphorus analysis.** For total phosphorus analysis in samples double acid digestion method was followed. For this samples were added with $(HNO_3)$, kept overnight, heated at (80–90°C) for approximately 1hr, until dense fumes subside. After solution cooled $HClO_4$ was added and heated 180–200°C until clear acid remains [24–31]. Free orthophosphate concentration in samples after digestion was determined following malachite green method and read under microplate reader at 620 nm [24–30]. Phosphorus in the bacterial culture after mineralization was also measured following malachite green method [24–26].

**Alkaline phosphatase activity.** The soil phosphatases enzyme in soil was determined by hydrolysis of an organic substrate p-nitrophenyl phosphate (pNPP) to p-nitrophenol. Sodium hydroxide is used to stop the assay and also to develop the yellow coloration (p-nitrophenol) for resultant analysis.

At the time of harvesting plant roots, samples were gently shaken, removed bulk soil and 0.3 g of soil was taken from each replicate of the treatment and diluted with 3 mL of sterile distilled water and incubated for one hour. Solution was vortexed, centrifuged at 14000 rpm, then 0.25 mL of this supernatant was taken and incubated with sodium acetate buffer containing 10 mM pNPP, re-incubated for 30 minutes at 30°C then 4 mL of 125 mM NaOH was added to allow for color not derived from p-nitrophenol released by phosphatase activity. Controls were set with each isolate sample by incubating 0.25 mL of water with 1 mL of pNPP and buffer and immediately adding 4ml of 125 mM NaOH to stop the reaction. Microplate was made with each sample replicates and production of pNPP from the color derived was determined at 405 nm [18–22].

**Root and shoot dry weight.** Roots and shoots were separated of each plant sample, dried in oven at 65°C for 48 hours, immediately weighed to reduce the possibility of moisture with precision balance with 0.001g accuracy [17].

### Soil and poultry manure pH and available P

To measure the pH and available P in soil and manure, suspension with 1gram solid and 10 ml deionized water was made Cui et al [23] Waldrip-Dail [10] pH was determined with pH meter. For available P (orthophosphate), while reagent 1 and 2 [24] was added in respective supernatant following malachite green method for determination of available phosphorus [17–31].

### Statistical analysis

Results were expressed as mean±standard deviation of different independent replicates. Data were analyzed using One-way and two-way ANOVA. Different letters were used to display significant differences. The differences between means were analyzed by factorial ANOVA followed by Tukey's HSD post-hoc test using Statistica 7.1 (StatSoft Inc., Tulsa, OK, USA). Normality was tested using the Kolmogorov Smirnov test to meet the assumptions of ANOVA.

## Results

### Net available phosphorus

The net water available P was significantly affected by predator treatment regardless of the amendment. The highest available P after the addition of both bacteria and nematodes in unamended treatment (40.9 µg/g was approximately twice higher than similar amended treatment 29.9 µg/g. As a whole $T_1$ (amended) treatment as shown in Fig 1 declined P availability 0.58 times as compared to unamended control. The unamended and amended bacteria only exhibited net P availability of 20.64 and 20.44 µg/g respectively. Which upon predation increased 2 times and 1.5 times net P respectively. The interactive effect of both amendment and predation on net available P was not significant.

### Alkaline phosphatase activity: At harvest, ALP activity

The ALP activity of unamended control soil (without PM, pH 5.8) was around 2.77 µmoles of pNPP/ml/hr while the addition of PM marginally reduced the ALP to 2.23 µmoles of pNPP/ ml/hr at pH 6.7. However, the unamended and amended predator treatment significantly increased ALP 6.47 µmoles of pNPP/ml/hr and 1.96 µmoles of pNPP/ml/hr respectively, compared to their respective controls. Amended bacteria only treatment (+bac) decreased ALP 1.65 µmoles of pNPP/ml/hr (0.8times) at pH 7.4 compared to its control and similar unamended treatment ALP increased (1.1 times) 2.91 µmoles of pNPP/ml/hr at pH 7.2 as described in Fig 2. According to ANOVA test, soil ALP tended to increase in predator treatment compared to its control but declined from similar unamended treatment. The ALP value was not significantly affected by the interaction of factors but significantly affected by predation (+bac+nmt) in the absence of PM.

### Effects of amendment and treatments on soil net available P Vs pH and Vs ALP

At pH 7.2 and 7.6 in amended or unamended soil, the net available P content changed from 22 to 41 µg/g, with highest amount found in unamended predation treatment at pH 7.56 and second highest in similar amended treatment (29.9 µg/g) at slight increased pH 7.6. Irrespective of the addition of PM, predation treatment increased P at similar pH 7.6 but significantly higher 41 µg/g of net P was found in unamended treatment ($T_0$). According to Fig 3A and 3B

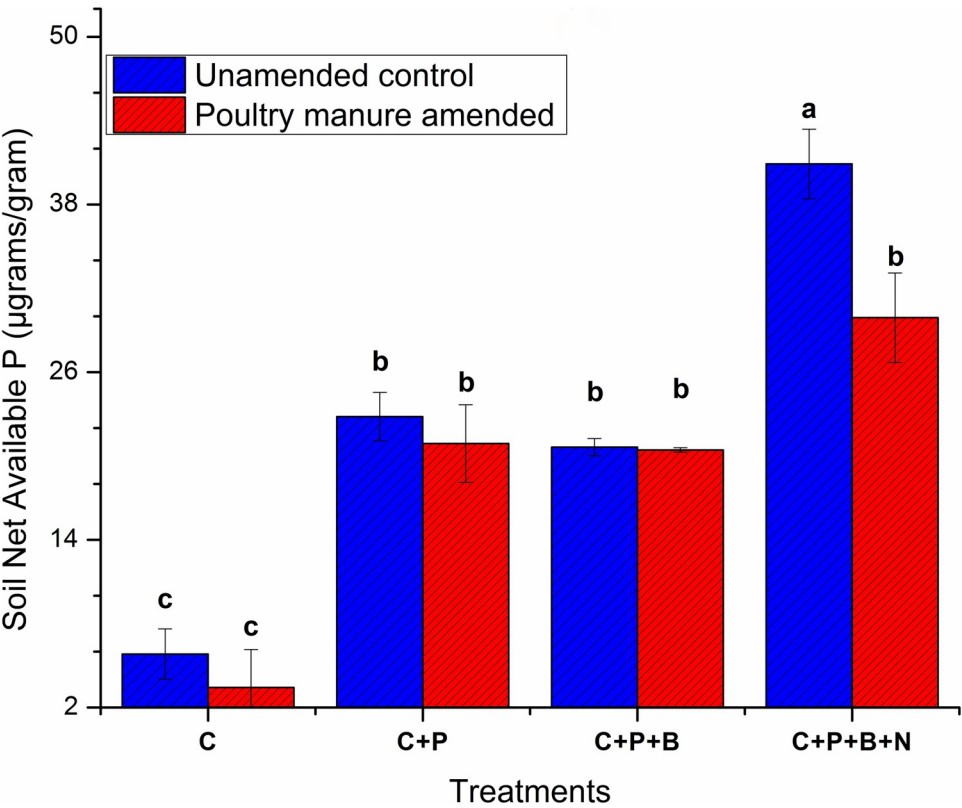

**Fig 1. Soil net P availability after 90 days of experiment.** C: Control, C+P: Control+wheat plant, C+P+B: Control +wheat plant+ bacterial consortia, C+P+B+N: Control+wheat plant+ bacterial consortia +nematodes: Bars are triplicate means of each treatment and standard deviation is shown by error bars. Significant difference is shown by letters (Tukey's HSD test, P ≤ 0.05).

the unamended and PM amended bacteria only treatment at pH 7.25 and 7.39 showed 20.9 and 20.4 μg/g, respectively.

Concerning the availability of phosphorus with respect to the enzyme, a lowest soil net P availability of 3.43μg/g (2.23 μmoles of pNPP/ml/hr) under PM amended control treatment was noticed. A significantly high ALP activity 6.47 μMoles/ml/hr with consequential net available P (40.9 μg/g) was observed in unamended (+bac+nmt) treatment. PM amended plant only treatment and predation treatment showed significantly similar ALP value of 1.97 and 1.96 μmoles of pNPP/ml/hr with 20.4 and 29.9 μg/g net P, respectively. While both unamended and amended bacteria only treatment (+bac consortia) retained the available P concentration to 21μg/g which increased upon nematodes inoculation. Overall, predator treatment, whether amended or unamended, increased ALP and net P compared to their respective controls, but unamended predator treatment compared to its similar amended treatment showed the significantly highest effect of predator treatment.

## Plant phosphorus concentration

Overall, Fig 4 showed a significantly higher P concentration was observed in wheat plant under +bac treatment 13490 μg/g DW/pot and + predator treatment 15404 μg/g DW/pot in the presence of PM. On the other hand, in the absence of manure amendment both inoculation treatments (+bac) and (+bac+nmt) showed the weakest influence on plant P concentration 4570.05 and 4369.01 μg/g DW/pot, respectively. The results of two-way ANOVA showed

**Fig 2. Soil alkaline phosphatase activity after 90 days of experiment.** C: Control, C+P: Control+wheat plant, C+P+B: control+wheat plant + bacterial consortia, C+P+B+N:Control+wheat plant+ bacterial consortia +nematodes: Bars are triplicate means of each treatment and standard deviation is shown by error bars. Significant difference is shown by letters (Tukey's HSD test, P ≤ 0.05).

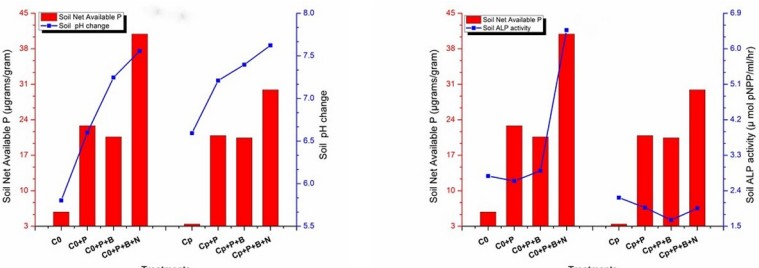

**Fig 3.** Soil net P availability versus a) soil pH change b) soil ALP activity with respect to treatments. C0: Unamended uninoculated control, Cp Poultry manure amended uninoculated control, C+P: Control+wheat plant, C+P+B: Control +wheat plant+ bacterial consortia, C+P+B+N: Control+wheat plant+ bacterial consortia +nematodes: Bars and lines are triplicate means of each treatment.

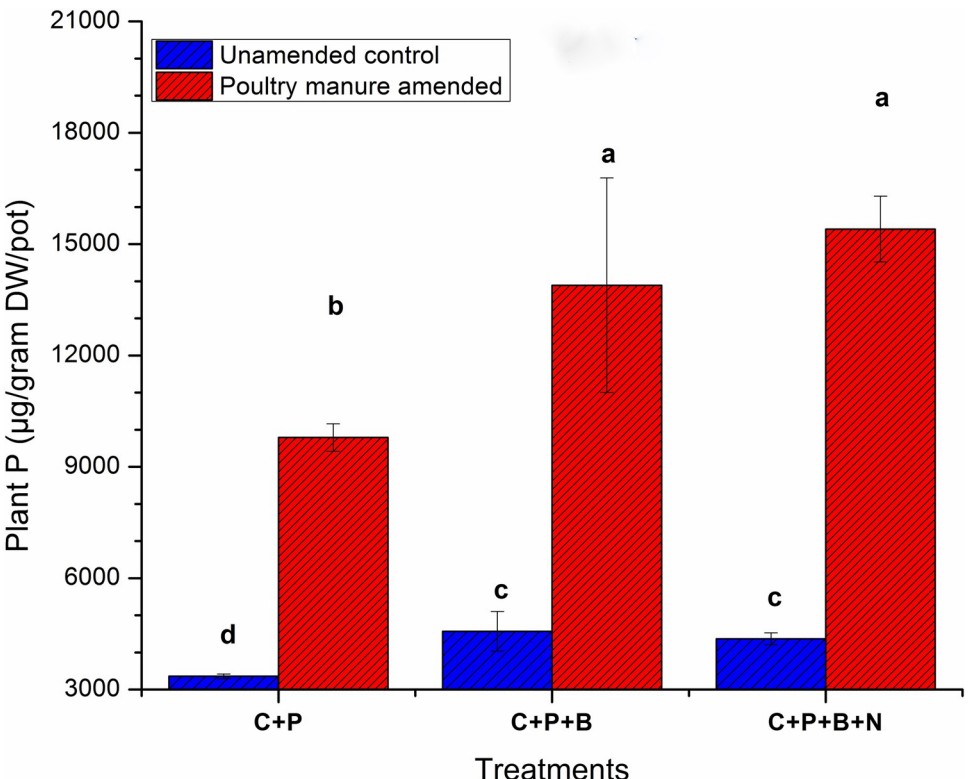

**Fig 4. Plant P concentration.** C: Control, C+P: Soil+wheat plant, C+P+B: Control+wheat plant+ bacterial consortia, C+P+B+N: Control+wheat plant+ bacterial consortia +nematodes: Bars are triplicate means of each treatment and standard deviation is shown by error bars. Significant difference is shown by letters (Tukey's HSD test, $P \leq 0.05$).

a significant interactive response (PM +bac+nmt) for plant P concentration. This result indicated the high dependence of nematodes mutualistic activity in the abiotic soil component i.e., PM substrate.

## Plant dry biomass

After 90 days of experiment, a significantly higher plant dry biomass was observed in PM amended treatment compared to the unamended treatment Fig 5. But no significant difference was observed on plant dry biomass between the treatments. The minimum of 358mg plant dry weight (DW) was observed in unamended uninoculated control while with the bacteria only (+bac) and predator treatment a non-significant increase of 434 to 441mg DW was observed. A significant increase was found with 620 mg DW of plant, in PM amended control than unamended control. Bacteria only (+bac+plant) treatment in the presence of PM exhibited the significantly higher plant DW (702 mg) compared to the similar unamended treatment. Plant DW was significantly affected by the interaction of treatments besides individual effect of both treatments.

## Presence of P mobilizing genes

Presence of genes responsible for orthophosphate production were performed in selected bacterial strains. phnX (147) phosphonoacetaldehyde hydrolase, phoN (159) acid phosphatase and Quinoprotein glucose dehydrogenase. PhnX gene was present in *Bacillus cereus*, *Bacillus thuringiensis and Bacillus subtillis* but in *Pseudomonas* it was under expressed. While phoN gene was expressed and gcd was successfully expressed in the *Bacillus subtillis*.

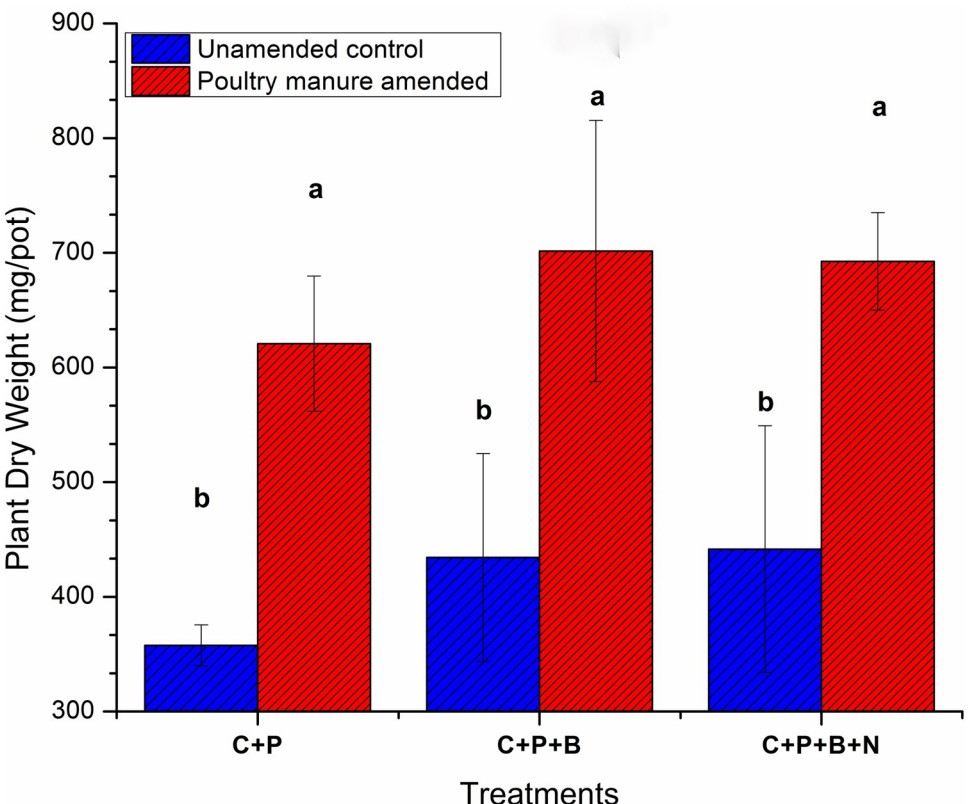

**Fig 5. Plant total dry weight.** C: Control, C+P: control+wheat plant, C+P+B: control+wheat plant+ bacterial consortia, C+P+B+N: Control+wheat plant+ bacterial consortia +nematodes: Bars are triplicate means of each treatment and standard deviation is shown by error bars. Significant difference is shown by letters (Tukey's HSD test, P ≤ 0.05).

## Discussion

### Microbial P dynamics changes in soil treatments

In this study the consequences of synergistic effect and trophic interactions among bacteria and their grazers on soil available P, plant P uptake and biomass were explored. The influence of interactions between poultry manure amendment representing input of P and trophic level was investigated, though, the trophic effects were noted both with and without PM addition but overall, results suggested that in this microcosm set-up trophic level (regardless of substrate limitation) can influence soil processes of P cycling.

During a three-month incubation, a significantly higher ALP and net soil P was noted compared to the similar PM amended treatment. No significant interaction between trophic level and PM addition was observed in this case Figs 1 and 2. Moreover, significantly similar net P concentration in the unamended sole plant and sole bacterial treatment and similar PM amended treatments was observed. The water available P concentration in sole plant treatment compared to their respective controls was increased by 3.9–6.2 times in both unamended and PM amended treatment, respectively. However, the unamended predator treatment increased 1.96–1.5 times more water available P.

Amended predator treatment improved plant dry weight and plant P concentration relative to unamended predator treatment while unamended treatment initially showed higher net available P and ALP due to nutrient starved condition. Alori et al [5] reported that water

available P (Pi in soil solution) increases only when the pool of microbial biomass decreases or when the microbial biomass becomes poorer in P, Under field conditions, the size of the microbial P pool can increase or decrease depending on the soil conditions e.g., soil humidity [5–33].It has also been suggested that the contribution of nematodes and other soil fauna become significant usually under the conditions of low nutrient availability [33,34].

This suggests that the presence of predator beyond bacterial treatment may exert a stronger influence on the microbial efficiency regardless of the addition of a labile substrate i.e., PM. Bünemann et al. [35] showed that P mineralization was independent of organic matter turnover. This implies an increased plant and microbial competition for nutrient which might be the result of P starvation which may have enhanced net P mineralization by 2 to 3 times. These results of significantly higher net P and ALP in similar unamended predator treatment is an indication of the higher net organic P mineralization supplemented with release of organic P from the bacterial biomass by nematode predation i.e., microfaunal stimulation of P mineralization via the microbial loop. Also, similar net P value in all sole plant and sole bacterial treatment may be related to the increase in P cycling internally as a P starvation phenomenon. Soil extracts when treated with phosphatase enzyme released significant amounts of orthophosphate which is in support to our net water available P and ALP results, indicating organic P mineralization [35]. It was investigated that reduction in the soil organic P is related to the high phosphatase activity [36–38]. Significantly positive effect in predator treatment without PM addition may be the response of nematodes predation as indicated by Trap et al [32]. As Bonkowski [36] showed that the effect of bacterivores nematodes on soil P availability for bio assimilation was triggered by the soil microbial and plant biomass. A significantly similar net P concentration in sole bacteria treatment with and without PM addition may be attributed to the initial low P availability in both treatments so bacteria retained the P in their biomass. The higher P availability in P fertilized soil was found by Sphon and widding [33] who reported that microorganisms retained P in their biomass in soil with low P availability, whereas in P fertilized soil P turnover time from microbial biomass was shorter and water extractable P concentration in soil increased. Bacteria mineralized P from soil organic matter and calculations by Irshad & Griffith [17–34] gave evidence that bacterivores grazing results in the release of this additional P. Using labelled bacteria Irshad et al [26] reported that grazing by nematodes and protozoa increased the availability of P to plants, originating from bacterial cells but also from soil organic matter. In a research study net nutrient (P and N) availability in microcosms study with bacteria alone and nematodes showed that, inorganic P was initially immobilized in bacterial biomass, with over half of the initial P returned after 65 d in both bacteria and predation treatment. The Sphon and widding [33] in their study demonstrated that the turnover time of microbial biomass nutrients varied from 10 to 160 days, depending upon the soil carbon and P inputs [38]. In a similar study with addition of C input in the form of grass clover residue inoculation of nematodes significantly enhanced the net P availability up to 23% and *Lolium perenne* plant biomass up to 9% compared to control without nematodes [32].

Net P (soil solution orthophosphate) declined by PM addition in bacteria and predation treatment in part, could be attributed to the underlying factor of increasing microbial biomass due to increased C availability which may have caused high incorporation of available P in their biomass [39]. In contrast, the work of Heuck et al [33–40] showed that more C was utilized by soil MOs from organic phosphate inputs in P deficient soil, this implies that organic P mineralization occurs as result of C requirement by microbes [33–41]. This mechanism might increase P availability to plants in soils where P is limiting for plants, but not for microorganisms. Overall unamended predator treatment enhance soil measured parameters i.e. Soil net P and ALP, while the amended predator treatment significantly influenced the plant P and DW, which apparently shows the the effect of amendment increased with time. Waldrip-Dail et al

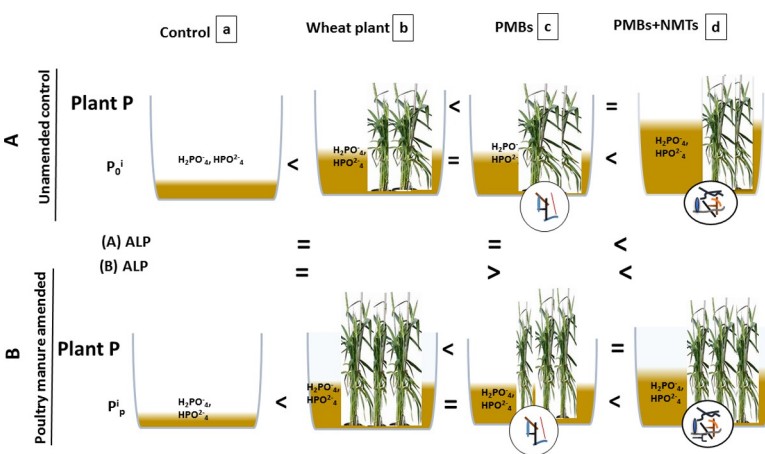

**Fig 6. Schematic diagram; presentation of comparison of individual and interactive influences of soil biological drivers and PM amendment.** On available P, plant P acquisition and ALP as a result of applied treatments as a, b c and d. soil biological interactions; a plant-bacteria-nematode priming established a beneficial nutrient loop. Arrows sign (<) indicate a significant increase, Sign of equality (=) shows non-significant difference ALP enzyme activity, available P and plant P are shown as ALP and $P_0i$. Ppi, respectively, Plant P, plants showing max height and plant number than other plants = high plant dry biomass, organisms in circles show inoculation of bacterial consortia/ nematodes and bacterial consortia.

[10] demonstrated a short-term initial increase in water extractable P by PM amendment, then decreased after 100 days controlled by immobilization-solubilization.

Significantly high ALP activity value in PM amended predation treatment Fig 6A (d) in current study may be associated with P starvation response as a result of P deficiency as study on organic P mineralization showed the reduction in organic P of soil when phosphatase activity was highest [37]. We observed 0.4 times decrease in the ALP activity in the PM amended predator treatment compared to the similar unamended treatment. While ALP activity in predator treatment compared to their respective controls decreased by 0.86 times and 1.96 times increase was observed with and without PM addition, respectively.

Alternately, saturation of soil nutrient status after amendments might have potentially inhibited or delayed the enzyme activity of soils in predation treatment with PM addition [9–28]. Knowing the same effect of predation on soil ALP and net P in the treatment without PM addition can reveal that how this enzyme is introduced because of P deficiency forecasting the more available phosphorus required for the plant uptake. Fertilization study further support this possibility, a positive correlation was found between inorganic P pool size and phosphatase. Alternately, P fertilization significantly decreased phosphatase activity. A study showed increase in the phosphatase activity in rhizosphere is associated with a depletion of soil organic P which an indicative of P mineralization [2–40]. While another study demonstrated a negative correlation of phosphatase activities with available P concentration in soil [16]. Heuck et al [40] reported that high P concentration do not warrant the high phosphatase activities, but this enzyme activity can be driven by the C requirement by the microbes. An increase in ALP activity was linked with organic P mineralization in the treatment where higher trophic levels are present (i.e., bacteria and their grazers) and organic manure addition is absent [26]. This result differs from Wan et al [2] who found high phosphatase activities in the P rich organic material amended soil and also depend on the soil pH and MB, besides, inverse relation has been found between enzyme activity and P availability after fertilization in various ecosystems. In support of our results, it was reported that fertilization decreased ALP activity rather ACP [16–33]. An overall effect of grazing on thenutrient (N and P) turnover estimated

by the meta-analysis was 30% of control [32]. Findings from another study suggest that when the supply P is sufficient phoD-harboring microorganisms immobilize P in biomass while mineralize organic P under P-poor condition by stimulating ALP activity, during which the rare taxa play an important role [39].

The overall impact of PM addition on the net available P and ALP was small even in the presence of biological drivers, while impact of biological drivers (predator treatment) was significant in this case and showed in unamended predator treatment Fig 6A unamended control.

## Linking crop yields and P acquisition with drivers of soil ecosystem functioning

Wheat plant dry biomass yield between amended and unamended experiment varied significantly with higher in PM amended and lower in unamended treatment. Bacteria and bacterivores mutually collaborate with significant effects on the root development, but it is uncertain that how much these processes control plant response [17–34]. Similarly, Trap et al [32] in meta-analysis and Gebermikael et al [19] showed higher plant P and plant biomass in the presence of nematodes and organic amendment and nematodes respectively. Soil fauna manage plant roots, root exudates and rhizosphere microbial processes interactions and therefore affects plant growth. According to a research report, carbon exudates from roots trigger microbial growth in the rhizosphere when no P is lost by exudation resulting in the organic P mineralization and assimilation into microbial biomass [42]. Considering that most studies reported the direct nutrient effects due to the microfaunal grazing on the rhizosphere bacteria i.e., the 'microbial loop on P mineralization and subsequently the increased growth and P transfer to the plants. We observed interaction between trophic level and PM addition similar to the changes observed for plant dry weight Fig 6B (c). Bunemann [35] reported that after organic amendment soil microbial immobilization of P prevent it from sorption and make it available for the plant uptake. We observed significant effect of PM addition with highest effect with sole bacterial treatment. The general substrate effect was primarily due to the increases in soil C pools and soil pH with PM addition because all the predator and microbe treatment without PM addition significantly reduced root P. There are increase as well as decrease in the microbial biomass as a result of predation, have been reported by [18,19] depending on the soil type, and carbon availability and various other factors [43], decline in the microbial abundance might be the reason for marginal decline in the root P uptake in predation treatment with PM addition.

Iqbal et al and Batool and Iqbal [7–12] in a study -of PSBs (Bacillus species) inoculation with the addition of P source reported more enhanced effect on root and shoot biomass and root P uptake rather P solubilization. In meta-analysis regarding the effect of bacterivores nematodes on the plant P and biomass their effect was reported up to 30% [32]. In the present study, it was observed 2.1 times more plant dry biomass in the sole bacteria treatment with the addition of PM compared to the similar unamended treatment. While predator treatment increases by 1.19 times more plant dry biomass compared to the similar unamended treatment. While shoot P was also significantly affected by the interaction of factors (drivers+substrate) with 2.5 times more shoot P acquisition in predator treatment with PM addition compared to similar unamended treatment. A strongly positive effect of Cephalobus sp. inoculated with Pseudomonas sp. or Burkholderia sp.was reported on root growth [17–44]. In a planted microcosms experiment shoot biomass and P uptake mostly increased in the presence of protozoa and nematode grazers showed strong evidence of microfaunal stimulation of P mineralization via the microbial loop as the main underlying mechanism [20]. There is

another evidence of positive influence of bacterivores nematode inoculation on higher P uptake and root length of rice amended with dolomite while higher root length wasn't linked with higher P uptake [44].

Here the individual impact of biological drivers and substrate was less pronounced but interaction of both appeared at plant growth stage as high dry plant biomass and plant P concentration Fig 6B (c & d).

## pH and microbial P dynamics

The influence of PM addition, microbial trophic interaction between PSBs and their grazer nematodes, and their interaction on the soil pH were investigated and it was found a significant available P content at pH 7.6 in the predation treatment without PM addition. Soil pH may be a good indicator for the organic P mineralization as a significant correlation between phoD gene and pH and P contents in soil were reported by Wan et al [2]. Wan et al Demonstrated pH as the main driver of high P contents, organic P mineralizing microflora [2–42]. As it was reported that available P content, organic P and ALP contents were correlated under various fertilization treatments Wan et al [5], found that pH strongly effect organic P mineralization which differ from [5] study demonstrated that organic phosphorus mineralizing enzyme activities decreased as Po increased. One explanation is that soil pH can directly influence Po-mineralizing-microflora, or interaction between them [5] thus affecting Po mineralization. The average PM amended plant P concentration was across both biological treatments was approximately thrice that of unamended similar treatments.

Assimilation and chemical factors such as precipitation and sorption causing P availability are sensitive to pH change [2–6]. There are diverse reports regarding the effect of pH change on plant P availability and uptake [41]. Organic P mineralization is also pH sensitive and increase at low pH [2–6].

Considering the system under poultry manure amendment and nematodes grazing on bacterial consortia could suggest that phosphorus release from the substrate by PMBs and their grazer is need dependent. But in phosphorus deficient soil PMBs activity were strong to make P available and in ALP production (P mineralization is high). The effect of substrate and predator is less comparable to unamended predator treatment, but predator effect was also significant compared to its relative control.

## Supporting information

**S1 Table. ALP activity and P mobilization values by 21 bacterial strains.**
(DOCX)

**S2 Table. ANOVA results showing the F and P values of significant differences between the applied treatments i.e. bioinoculation and amendment and their interaction.**
(DOCX)

**S1 Raw images. Unadjusted and uncropped images underlying all gel figures.**
(PDF)

**S1 File. Excel file containing raw data obtained from analysis.**
(XLSX)

## Acknowledgments

We are thankful to other lab colleagues and research students who helped us to manage the experiment during COVID-19 lock down period.

## Author Contributions

**Conceptualization:** Xiaoyan Tang, Usman Irshad.

**Data curation:** Saba Ahmed, Rafiq Ahmad.

**Formal analysis:** Rafiq Ahmad, Usman Irshad.

**Funding acquisition:** Usman Irshad.

**Investigation:** Saba Ahmed.

**Methodology:** Saba Ahmed, Nadeem Iqbal, Xiaoyan Tang.

**Project administration:** Usman Irshad.

**Resources:** Xiaoyan Tang.

**Supervision:** Usman Irshad.

**Writing – original draft:** Saba Ahmed.

**Writing – review & editing:** Saba Ahmed, Rafiq Ahmad, Muhammad Irshad, Usman Irshad.

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
