## [Decision Letter · Decision Letter 0]

28 Oct 2021

PONE-D-21-30854Organic Amendment plus inoculum drivers: who drives more P nutrition for wheat plant fitness in small duration soil experimentPLOS ONE

Dear Dr. Irshad,

Thank you for submitting your manuscript to PLOS ONE. After careful consideration, we feel that it has merit but does not fully meet PLOS ONE’s publication criteria as it currently stands. Therefore, we invite you to submit a revised version of the manuscript that addresses the points raised during the review process.

We look forward to receiving your revised manuscript.

Kind regards,

Academic Editor

PLOS ONE

Journal Requirements:

 [The research project was supported by a grant from the Higher Education Commission of Pakistan via an NRPU grant No: 20-3655/R&D/HEC/14/400.]

[The research project was supported by a grant from the Higher Education Commission of Pakistan via an NRPU grant No: 20-3655/R&D/HEC/14/400.]

 [The research project was supported by a grant from the Higher Education Commission of Pakistan via an NRPU grant No: 20-3655/R&D/HEC/14/400.]

7. PLOS requires an ORCID iD for the corresponding author in Editorial Manager on papers submitted after December 6th, 2016. Please ensure that you have an ORCID iD and that it is validated in Editorial Manager. To do this, go to ‘Update my Information’ (in the upper left-hand corner of the main menu), and click on the Fetch/Validate link next to the ORCID field. This will take you to the ORCID site and allow you to create a new iD or authenticate a pre-existing iD in Editorial Manager. Please see the following video for instructions on linking an ORCID iD to your Editorial Manager account: https://www.youtube.com/watch?v=_xcclfuvtxQ.

8. Please ensure that you refer to Figures 4-8 in your text as, if accepted, production will need this reference to link the reader to the figure.

Additional Editor Comments:

Use the ‘title case’ in title Or replace “A” with “a” in amendment…

Figures’ overall quality is not good; they are not clear to be read; please follow the journal’s guidelines. There are too many figures and some will not affect the manuscript-quality if excluded e.g. the figure-1 and Figure-9 don’t provide much flesh to the ms, fig. 2, 3, 4, and 5 have some redundancy and can better be re-represented. Fig. 6 and 7 can also be presented as one figure with A and B panels.

Figure 4: the legend says – ‘soil net pH-change’ while the data shows about pH values. pH-change means 1-fold, 2-fold, 3-fold etc. while the author seem to mean the actual soil pH and not the net-changes; better to modify accordingly to improve clarity.

What is the purpose of Fig. 2 and 3 in the presence of Fig. 5?

Figure 6/7 root/shoot P and not root/shoot P-uptake; I think the uptake means the mechanistic accomplishment of plant from one medium (soil) into the other (plant-body) via some transporter systems while the authors’ work indicates the values/concentrations present in root/shoot. Please clarify.

Figure 8: Plant total dry weight per pot; how did you normalize this? Were there same number of plants in each pot for all treatments? What was the normalization/standardization of plant health before the treatment application?

Figure 9: For molecular gene analysis: on the basis of PCR and gel electrophoresis bands, how can you claim the gene expression?? Manuscript’s sub-section “Successful identification of organic P mineralizing genes” looks inappropriate; the gel pictures simply mean the presence or absence of the potential genes and is not the gene expression. Neither were the amplified pcr-products sequenced, so cannot be said as identification either. Moreover, the authors used 20-30 years old primer systems; it would be surprising to know if there is not primer-improvement (for better specificity) during these years?? And, lastly, what is the role of Quinoprotein glucose dehydrogenase in organic P-mineralization?

Figure 10 is quite complicated, and its legend is even more. As this diagram is the summary of their whole work, the authors should improve it to make it comprehensive as well as clear to understand.

Table-2: replace bacteria name codes with ‘bacterial codes’. What does the column 3 and 5 mean?

As per Table-3, total soil P and soil organic P are the same that means this soil does not have any inorganic P; what do the authors say in this regard??

As per ms data, Soil net available P and soil ALP is always less in manure-amended treatments than the control treatments (Fig 2,3,4,5) ; does it mean that the organic farm manure application are not good while the farmers are frequently using this strategy as fertilization?? On contrary, the authors found enhanced root/shoot P in manure-amended treatments than the control treatments (Fig 6, 7). I recommend the authors to validate these observations and to make the appropriate discussion accordingly.

Reviewers' comments:

Reviewer's Responses to Questions

**Comments to the Author**

1. Is the manuscript technically sound, and do the data support the conclusions?

Reviewer #1: Yes

Reviewer #2: Partly

2. Has the statistical analysis been performed appropriately and rigorously? 

Reviewer #1: Yes

Reviewer #2: Yes

3. Have the authors made all data underlying the findings in their manuscript fully available?

Reviewer #1: Yes

Reviewer #2: Yes

4. Is the manuscript presented in an intelligible fashion and written in standard English?

Reviewer #1: Yes

Reviewer #2: Yes

5. Review Comments to the Author

Reviewer #1: The current study examined the potential effects of co-inoculation of phosphorus solubilizing bacteria and their grazer nematodes on soil P availability in presence of poultry manure. The subject of this article is coherent with the scope of Plos ONE. Given the depletion of the limited P-fertilizer resources, scientific understanding the interaction of soil microfauna to improve the soil P fertility is important. The manuscript is well written. However, the manuscript requires revisions before it could be accepted for publication.

Specific comments:

Need to define first PSB.

L122: “A synthetic solid medium” is the starting sentence of L123.

Material and method

A huge amount of information is presented in the mat&met section. Some sections could be presented as a supplementary information to simplify the paper.

L110-L121: The soil properties should be presented with emphasis on the soil type (P-deficient or not, alkaline or acidic soil).

L136: PKO???

L168-L169: What criteria did you choose to select the 6 efficient strains of the 21 based on pH, ALP activity and P mineralization? Please report the results even as a supplementary information.

L287-L289: Can you provide the characteristic/succinct history of the field where soil was sampled (e.g. previous crop).

L302-L3016: Summarize the tested treatments of the experimental design in table to better help readers. Specify the duration of the experiment. How did you collected the soil samples for subsequent analyses (rhizosphere or bulk for each pot).

In statistical analysis section: provide sufficient information for data handling: software, the tested treatment with the statistical approach used (here one and two-way anova). The test of normality of the data and the homogeneity of variance.

Results:

L376-L381: Give the Anova result either in the text or in a separate table. Describe the T1 treatment here. How did you calculate the soil net P availability in Fig 2? May be it’s better to add this information in mat&met section.

I think it’s better to present the result for each tested parameters first before going in a specific description in your text: bacteria treatment, Nematode treatment, Poultry treatment, and interactions, and should be supported by Anova results. The same also with ALP results in L388-L395, and with pH and soil net available P (L402-L406).

It is interesting to see the different response of soil net available P with soil net pH. Increasing soil pH from 5.75 to 7.5 (control to Control+wheat plant+ bacterial consortia +nematodes) increased soil P from 0.0.4 to 4 µg.g-1) in unamended treatment , while increasing soil pH from 6.5 to 7.5 increased soil P from 0.0.2 to 2.8 µg.g-1). Focusing in nematode treatment impact, how about the response of increasing soil P to soil pH between the two treatments C+P+B vs C+P+B+N with or without PM addition?

The same remarks with the net P liberation vs ALP activity.

The clear effect of Nematode under PM amendment or no organic amendment could be describe and general conclusion can be drawn from this.

I suggest that authors presents the correlation between the measured parameters as ALP, pH, soil P, and plant biomass and P uptake.

Discussion:

I suggest to clearly state the nematode predator effect in presence of PM and without PM on the measured parameters before discussing the underlying mechanism in L507.

L580: Fig 9???

L584: Fig 8 instead of Fig 9.

In the section “Microbial P dynamics changes in soil treatments”, L507-L555, authors stated that the predation treatment declined Net P and ALP activity with PM addition in contrast with those without PM amendment; this is related to the immobilization of available P in microbial biomass. Does it suggest that predator effect is OM-dependent in soil? Through which mechanism?

Total P uptake (P root + P shoot) could be added to clearly show the contrasting effect of treatment. Could you try to calculate the P uptake efficiency (PUPE as the ability of crops to uptake P from the soil) = P uptake/soil P available (Moll et al. (1982)) to verify the predator , PM treatment effects.

Do you have any results on soil available P & ALP activity measurement with time (incubation of microcosm study)? This could be interested to be related to the higher plant P uptake & biomass in PM amendment. Is it possible that after high P uptake, soil could be depleted in available P? Because here soil P & ALP seems to be negatively correlated with plant P uptake.

I consider the data from different experiments including strains screening and microcosm experiment should be highly valuable and hopefully presented. I would like to encourage the authors to present clearly the data analysis in the result section and to resubmit the revised manuscript.

I hope the specific comments may also help authors for the revision.

Reviewer #2: # Review report: Ahmed et al.,

The submitted manuscript with the entitled "Organic Amendment plus inoculum drivers: who drives more P nutrition for wheat plant fitness in small duration soil experiment" is a very nice research article with up-to-date information submitted by Dr. Usman Irshad as a corresponding author. The research article describes the importance of biological drivers that significantly enhanced the soil ALP and available P and thus enhanced dry biomass and P uptake. However, reviewer has few suggestions to improve the impact of research article to the readers. Comments, and suggestion to the author are the following:

Title: Organic Amendment plus inoculum drivers: who drives more P nutrition for wheat plant

fitness in small duration soil experiment

The title of this manuscript does not match the results. Kindly, check it and reframe it accordingly.

Abstract: It is too lengthy, and sentences are repeated. Kindly make it short and put the main findings here.

Line 30-What is WEP?

Note-Key word are missing.

Introduction: The introduction of this manuscript is up to date. However, the authors used much time etc. Instead of etc. put more information that is required to strengthen your manuscript. Kindly do not use etc words.

Line 37- act as major sink and source (What does it mean?) of key abiotic components such as carbon

Line 42- development and growth of plant (Reference?). Agriculture soils are strongly depleted in available P for plants (How??), despite being virtually sufficient in total P 50-65% [3]

Line 53- Organic amendments of??

Line 55-Biomass of??

Line 56- other essential ecological functions such as?

Line 61- large amount of P in their biomass (Kindly write the proper sentences. I did not get the point here)

Line 64- Depending on the soil and fertilizer input the release of P from microbial biomass-How??

Question 1: Why did you choose only poultry manure?

Question 2: What was the concentration of P in soil before adding PM? And what was the content of P in PM?

Question 3: How did you decide the experiments duration for 3-months? Kindly, mention the specific reason.

Line 90-91: These microbial growths sequester large amounts of available nutrients such as P, C and N, e.g., nematodes while grazing sequester 50–70% of the prey carbon for biomass production [19]. Kindly rewrite the whole sentence to make it clear for audience.

Materials and methods:

Comments-I would like to suggest to the authors kindly make it short and crisp with only required information. Materials and methods are too lengthy as compared to results sections. You can shift unnecessary part of this section to the supplementary sections.

Note-Kindly make the italic all the scientific names.

Line 113- The soil samples were collected from the upper soil layer (depth 113 0-15cm) in triplicates, A substantial P turnover occur in the surface layers of soil and microbial number reduce along soil depth [21]-Why did you write this statement in materials and methods section?

Line 119- while the remaining portion of soil was mixed with?

Line 118- Isolation of phosphorus solubilizing bacteria from? And how did you isolate?

Line 120- soil initial pH, water available phosphorus and total phosphorus. All determinations and results were done in triplicate (Kindly make it clear). Line 122- Isolation, screening and identification of PSBs on A (Why this is capital?) synthetic solid medium

Question 4-How did you control the growth of other microorganisms?

Line 165- with each isolate sample by incubating 0.25 mL of water with 1 mL (Space) of pNPP and buffer and

Line 166- immediately adding 4 mL (space) of 125 (Space) mM NaOH to stop the reaction.

Line 169- 6 efficient strains out of 21 were selected for further analys (remove extra space)es. This statement is not supported by your table no-2.

Line 177- Phosphorus mineralizing gene expression using q-PCR: In bacteria phoN, phnX gene (full form?) encoding enzymes are responsible for organic P mineralization and gcd (full form?) for inorganic P solubilization i.e.,

Note-Figure2-Kindly prepare a new diagram with proper formatting. You must have to keep the size of figures 300 dpi.

Line 227- The synergistic activity by combination of 6, 4, 3 and 2 of 6 or 4?? selected isolates

with high available

Figure 1: This is not providing a clear message to the readers; therefore, I would like to ask to make a table for figure 1.

Results:

Note-Kindly, prepare all the figures with proper formation to keep at least 300 dpi size.

Figure 9-You should replace this figure with other. As I can see clearly shadow instead of bands. I am not able to see the proper bands.

Note-In each results heading, you should write outcome of the results (kind of 1 line result). For example, do not write Plant dry biomass. You can write treatment of X and Y enhanced dry biomass.

Note-Conclusion of this manuscript is missing. Kindly write it.

Once you address all the comments and queries, after that, I will recommend this manuscript for further process.

6. PLOS authors have the option to publish the peer review history of their article (what does this mean?). If published, this will include your full peer review and any attached files.

Reviewer #1: No

Reviewer #2: No

---

## [Author Response · Author response to Decision Letter 0]

2 Mar 2022

Responses to Reviewers/Editors Comments

First of all, we are thankful to Editors and reviewers for valuable suggestion and correction in our manuscript. We incorporated all the suggestion and comments and please find the detail of each improvement here. 

General Comments

Q1. Use the ‘title case’ in title Or replace “A” with “a” in amendment…

Answer: Thanks for highlighting this issue with title. We have incorporated your suggestion as

‘Organic amendment plus inoculum drivers: who drives more P nutrition for wheat plant fitness in small duration soil experiment’.

Q2. Figures’ overall quality is not good; they are not clear to be read; please follow the journal’s guidelines. There are too many figures and some will not affect the manuscript-quality if excluded e.g. the figure-1 and Figure-9 don’t provide much flesh to the ms, fig. 2, 3, 4, and 5 have some redundancy and can better be re-represented. Fig. 6 and 7 can also be presented as one figure with A and B panels.

Figure 4: the legend says – ‘soil net pH-change’ while the data shows about pH values. pH-change means 1-fold, 2-fold, 3-fold etc. while the author seem to mean the actual soil pH and not the net-changes; better to modify accordingly to improve clarity.

Answer: We agree with editors’ observation and we revised all figures as per Journals criteria (saved more than 600 dpi). In current version figures numbers reduced from 10 to 6, further as per Editor’s suggestion figure 1 and 9 removed from the revised manuscript. Figures 2, 3, 4 and 5 are represented in high quality as fig. 1, 2, 3a & 3b in current revised manuscript. Figures 6 and 7 are combined together as one figure (fig. 4 now) as per reviewers and Editor remarks.

Figure 4 now represented as figure 3a modified regarding pH change. In current version it is showing actual pH change rather than net pH change as suggested by Editor. 

Q3. What is the purpose of Fig. 2 and 3 in the presence of Fig. 5?

Answer: We have reduced and changed the presented figures by taking in account all the comments. We still feel that figures 2 and 3 representing only one parameter data respectively in older version. While figure 5 currently presented as fig. 3b representing the data of two parameters which are dependent on each other. We tried to show the trend of these parameters in graph that how two parameters respond against applied amendments and treatments. We wish to show their trends either in case of amended experiments or unamended together putting both parameters.

Q4. Figure 6/7 root/shoot P and not root/shoot P-uptake; I think the uptake means the mechanistic accomplishment of plant from one medium (soil) into the other (plant-body) via some transporter systems while the authors’ work indicates the values/concentrations present in root/shoot. Please clarify.

Answer: Thanks for nice suggestion we strongly agree with Editor’s suggestion and changed our statement from root/shoot P uptake to ‘Total Plant P’ in graphs. 

Q5. Figure 8: Plant total dry weight per pot; how did you normalize this? Were there same number of plants in each pot for all treatments? What was the normalization/standardization of plant health before the treatment application?

Answer: Yes sir, we normalized by same numbers of plants per pot after germination by pulling out extra wheat seedlings. As per plant health concerns, the almost similar height and number of leaves of different plants were measured and considered as their health status before treatment application. Further visual observation based on color were monitored and found approximately similar in all pots. 

Q6. Figure 9: For molecular gene analysis: on the basis of PCR and gel electrophoresis bands, how can you claim the gene expression?? Manuscript’s sub-section “Successful identification of organic P mineralizing genes” looks inappropriate; the gel pictures simply mean the presence or absence of the potential genes and is not the gene expression. Neither were the amplified pcr-products sequenced, so cannot be said as identification either. Moreover, the authors used 20-30 years old primer systems; it would be surprising to know if there is not primer-improvement (for better specificity) during these years?? And, lastly, what is the role of Quinoprotein glucose dehydrogenase in organic P-mineralization?

Answer: Yes, we do agree with reviewer that we did not study gene expression, we studied the presence and absence of genes through PCR. So, we corrected the mentioned results as per Editor suggestion in revised manuscript. As the studied bacteria are already sequenced and we designed specific primers for studied genes, therefore, sequencing is not necessary for identification. Moreover, we checked the primers sequence in studied genes and these primers sequences were present and these are already optimized, that’s why, we used these primers as well as these primers are still in practice, and we took help from a recent paper published in ‘Journal of Microbiological Methods’ 5 years ago. The presence of gcd gene in our setup indicated that it’s not only P-mineralization phenomenon happening in experiment. The other related P fixation after mineralization phenomenon also happening in same system, so as per literature, people use this gcd as an indication of microbial P turnover mechanism. Further our inoculated consortia possess not only mineralizers but also P solubilizers having gcd genes. 

Q7. Figure 10 is quite complicated, and its legend is even more. As this diagram is the summary of their whole work, the authors should improve it to make it comprehensive as well as clear to understand.

Answer: We have simplified the figure 10 as suggested which is represented as figure 6 in revised manuscript. We tried to remove the data presented and highlighted the mechanism by the use of arrow signs. 

Q8. Table-2: replace bacteria name codes with ‘bacterial codes’. What does the column 3 and 5 mean?

As per Table-3, total soil P and soil organic P are the same that means this soil does not have any inorganic P; what do the authors say in this regard??

Answer: The bacterial names were replaced with bacterial codes in revised manuscript. In Table 2 column 3 indicates alkaline phosphatase produced by bacteria when grown in liquid culture and column 5 indicated the Pi release in same medium from organic P sources (phytate sodium salt). We have completed the headings and now they are more explicit in table 1 of revised manuscript. Thanks for indicating this technical point we have provided the standard deviation data with means in current manuscript which showed huge variation in available/water soluble P in soil. Further we have changed the organic P heading of the table as ‘unavailable P’.

Q9. As per ms data, Soil net available P and soil ALP is always less in manure-amended treatments than the control treatments (Fig 2,3,4,5); does it mean that the organic farm manure application are not good while the farmers are frequently using this strategy as fertilization?? On contrary, the authors found enhanced root/shoot P in manure-amended treatments than the control treatments (Fig 6, 7). I recommend the authors to validate these observations and to make the appropriate discussion accordingly.

Answer: Thank you for the interest. Your consideration and valuable advice made a huge improvement in our research paper. Yes, as it is evident from research literature that ALP activity by MOs is high under the nutrient starved condition. Furthermore, manure has many binding sites which upon application in soil convert the available P into complex compound (unavailable P) therefore the release of Pi in soil is need based. Farmers are using livestock manure but biodiversity enrichment is very necessary, so the application of biological drivers is key point to better get a positive output from added manure. We were not able to sample the experiment on temporal basis due to the less soil quantity used in our experimental system. That might show us the point where we have prominent changes in soil available Pi before it gets fixed with manure. There could be more Pi availability on the course of experimental duration which released efficiently only upon need, so manure applied soil contributed more P to plant compared to unamended treatment.

 

Reviewers' comments:

Reviewer's Responses to Questions

Comments to the Author

1. Is the manuscript technically sound, and do the data support the conclusions?

Reviewer #1: Yes

Reviewer #2: Partly

Answer: Thanks for your valuable and encouraging feedback

2. Has the statistical analysis been performed appropriately and rigorously?

Reviewer #1: Yes

Reviewer #2: Yes

Answer: We appreciate your response and consideration.

3. Have the authors made all data underlying the findings in their manuscript fully available?

Reviewer #1: Yes

Reviewer #2: Yes

Answer: Thanks for your positive feedback and encouragement.

4. Is the manuscript presented in an intelligible fashion and written in standard English?

Reviewer #1: Yes

Reviewer #2: Yes

Answer: Thanks for your attention and positive feedback.

5. Review Comments to the Author

Reviewer #1: The current study examined the potential effects of co-inoculation of phosphorus solubilizing bacteria and their grazer nematodes on soil P availability in presence of poultry manure. The subject of this article is coherent with the scope of Plos ONE. Given the depletion of the limited P-fertilizer resources, scientific understanding the interaction of soil microfauna to improve the soil P fertility is important. The manuscript is well written. However, the manuscript requires revisions before it could be accepted for publication.

Specific comments:

Q1. Need to define first PSB.

Answer: Thank you to point out this deficiency. We have defined and expanded the abbreviated “PSBs” (as phosphate solubilizing bacteria) in the revised manuscript in introduction section from line 70 to line 73 and abstract section in line 14. 

Q2. L122: “A synthetic solid medium” is the starting sentence of L123.

Answer: Dear reviewer thank you again for highlighting the improvements we need to do. We have corrected this formatting mistake presented in L120 of revised manuscript.

Q2. Material and method A huge amount of information is presented in the mat&met section. Some sections could be presented as a supplementary information to simplify the paper.

Answer: Respected reviewer thank you for valuable advice. We have worked on mat and met section and squeezed certain sections especially isolation, experimental design (231-243), compatibility test (196-203), phosphorus analysis (259-266), ALP (267-279) and DNA extraction (167-185). Now it is squeezed (previous version line 110 to 372) to revised version from line 111 to line 294. 

Q3. L110-L121: The soil properties should be presented with emphasis on the soil type (P-deficient or not, alkaline or acidic soil).

Answer: The soil used in experiment was loamy clay with deficiency in available plant P possessing slight alkaline properties. The same presented with previous crop history of soil in section sampling and processing of soil and poultry manure of Materials and Methods.

Q4. L136: PKO???

Answer: Dear reviewer Thank you for highlighting the required improvement. In the revised manuscript the abbreviated PKO is now represented as Pikovskaya. 

Q5. L168-L169: What criteria did you choose to select the 6 efficient strains of the 21 based on pH, ALP activity and P mineralization? Please report the results even as a supplementary information.

Answer: Respected reviewer thank you for your valuable comment.

We have selected the 4 efficient strains out of 21 on the basis of maximum ALP and Pi release from phytate source when grown in media with phytate as sole P source. Then these strains further tested for compatibility against each other to make consortia. Results regarding ALP and Pi release of 21 strains is provided as supplementary data as S table 3. 

Q.4. L287-L289: Can you provide the characteristic/succinct history of the field where soil was sampled (e.g. previous crop).

Answer: Dear reviewer after your valuable advice we have incorporated the desired information in material and method section in L236.

Q5. L302-L3016: Summarize the tested treatments of the experimental design in table to better help readers. Specify the duration of the experiment. How did you collected the soil samples for subsequent analyses (rhizosphere or bulk for each pot).

Answer: Respected reviewer thanks for the suggestion. We have summarized and simplified the experimental design test treatments in mat and met section; L232 to 257 in revised manuscript previous 288-321. We have specified the duration of the experiment as 90 days in abstract as well. For soil initial analyses we have collected bulk soil, while for final analyses rhizosphere samples were taken.

Q5. In statistical analysis section: provide sufficient information for data handling: software, the tested treatment with the statistical approach used (here one and two-way anova). The test of normality of the data and the homogeneity of variance.

Answer: Unless otherwise stated, the results are given as mean ± standard deviation (n 3). The differences between means were analyzed by factorial ANOVA followed by Tukey’s HSD post-hoc test using Statistica 7.1 (StatSoft Inc., Tulsa, OK, USA). Normality was tested using the Kolmogorov Smirnov test to meet the assumptions of ANOVA.

Results:

Q6. L376-L381: Give the Anova result either in the text or in a separate table. Describe the T1 treatment here. How did you calculate the soil net P availability in Fig 2? May be it’s better to add this information in mat&met section.

Answer: Dear reviewer thanks for your advice and suggestion. We have provided the ANOVA results as supplementary data as Supplementary Table 2. We have calculated the soil net P availability by subsequent subtraction of inoculation treatment from previously un inoculated treatment, which was due to the effect of inoculation. T1 treatment is set after poultry manure addition at 5% by weight compared to T0 without amendment. This information is also added in revised manuscript where found appropriate. 

Q7. I think it’s better to present the result for each tested parameters first before going in a specific description in your text: bacteria treatment, Nematode treatment, Poultry treatment, and interactions, and should be supported by Anova results. The same also with ALP results in L388-L395, and with pH and soil net available P (L402-L406).

Answer: Dear reviewer we thank you for your valuable advice and suggestion. In every section of the result where we got significant results, we highlighted it by proper statement; each subheadings of result section.

Q8. It is interesting to see the different response of soil net available P with soil net pH. Increasing soil pH from 5.75 to 7.5 (control to Control+wheat plant+ bacterial consortia +nematodes) increased soil P from 0.0.4 to 4 µg.g-1) in unamended treatment , while increasing soil pH from 6.5 to 7.5 increased soil P from 0.0.2 to 2.8 µg.g-1). Focusing in nematode treatment impact, how about the response of increasing soil P to soil pH between the two treatments C+P+B vs C+P+B+N with or without PM addition? The same remarks with the net P liberation vs ALP activity.

Answer: Thanks for the encouragement. Yes, the net soil P change from sole bacteria treatment to predator treatment in both amended and unamended treatments. This result is stated in the result section in revised manuscript. Similar increasing trend was observed in case of soil ALP, before and after nematode inoculation in both unamended and amended bacteria only treatments.

Q9. The clear effect of Nematode under PM amendment or no organic amendment could be describe and general conclusion can be drawn from this.

Answer: Dear reviewer thank you again for your valuable comment. Yes predator treatment affect was observed in both amended and unamended treatment. The predator effect was more noticeable at start, but plant health shows the pronounced amendment effect.

 Q10. I suggest that authors presents the correlation between the measured parameters as ALP, pH, soil P, and plant biomass and P uptake.

Answer: Dear reviewer thank you very much for your comment and valuable suggestion. We have tried a correlation analyses between the tested parameters, but there was less correlation found due to more treatments. But we tried to show the link between to measured parameters in the form of trend as shown in figure 3a and 3b in revised manuscript. This correlation analysis could have been possible if the duration of the experiment was longer. 

Discussion:

Q11. I suggest to clearly state the nematode predator effect in presence of PM and without PM on the measured parameters before discussing the underlying mechanism in L507.

Answer: Respected reviewer thank you for the valuable suggestion. Yes, we have clearly mentioned the predator effect with and without amendment in discussion.

Q12. L580: Fig 9???

Q13. L584: Fig 8 instead of Fig 9.

Answer: Dear reviewer thank you for highlighting the required corrections. We have provided the figure reference with correct number in revised manuscript in discussion section.

Q13. In the section “Microbial P dynamics changes in soil treatments”, L507-L555, authors stated that the predation treatment declined Net P and ALP activity with PM addition in contrast with those without PM amendment; this is related to the immobilization of available P in microbial biomass. Does it suggest that predator effect is OM-dependent in soil? Through which mechanism?

Answer: Thank you for the valuable comment. Yes, it is stated the decrease in net available P in predator treatment with amendment compared to similar unamended treatment. It is due to that manure has many binding sites which upon application in soil convert the available P into complex compound (unavailable P) therefore the release of Pi in soil is need based. Yes, and for second part the predation effects depend upon microbial biomass which is dependent upon amendment. 

That is why predation is indirectly OM dependent.

 Q14. Total P uptake (P root + P shoot) could be added to clearly show the contrasting effect of treatment. Could you try to calculate the P uptake efficiency (PUPE as the ability of crops to uptake P from the soil) = P uptake/soil P available (Moll et al. (1982)) to verify the predator , PM treatment effects.

Do you have any results on soil available P & ALP activity measurement with time (incubation of microcosm study)? This could be interested to be related to the higher plant P uptake & biomass in PM amendment. Is it possible that after high P uptake, soil could be depleted in available P? Because here soil P & ALP seems to be negatively correlated with plant P uptake.

Answer: Respected reviewer thank you very much for the valuable suggestions. 

We tried to calculate PUPE according to your suggestion. The data we obtained showed significant increase in PUPE on amended site of inoculation treatments. But as per Editor suggestion we need to show P concentration in plants not uptake by plants so in revised manuscript we are providing Total plant P not plant P uptake.

Due to the short duration of the experiment and less soil volume used we were only able to analyze the soil ALP and P at initial and final stages.

Yes, it is possible. We have observed high plant P concentration in amended treatment than its comparable unamended treatment with less net available P. 

I consider the data from different experiments including strains screening and microcosm experiment should be highly valuable and hopefully presented. I would like to encourage the authors to present clearly the data analysis in the result section and to resubmit the revised manuscript.

I hope the specific comments may also help authors for the revision.

Answer: Thanks for your valuable feedback. We really appreciate your contribution for improvement of manuscript. 

 

Reviewer #2: # Review report: Ahmed et al.,

The submitted manuscript with the entitled ‘Organic Amendment plus inoculum drivers: who drives more P nutrition for wheat plant fitness in small duration soil experiment’ is a very nice research article with up-to-date information submitted by Dr. Usman Irshad as a corresponding author. The research article describes the importance of biological drivers that significantly enhanced the soil ALP and available P and thus enhanced dry biomass and P uptake. However, reviewer has few suggestions to improve the impact of research article to the readers. Comments, and suggestion to the author are the following:

Title: Organic Amendment plus inoculum drivers: who drives more P nutrition for wheat plant fitness in small duration soil experiment

 The title of this manuscript does not match the results. Kindly, check it and reframe it accordingly.

Answer: Dear reviewer thank you for the comment and suggestion. We have tried to incorporate all the suggestion from the respected reviewers. Now the manuscript contents reflect the title more closely in revised version. 

Abstract: It is too lengthy, and sentences are repeated. Kindly make it short and put the main findings here.

Answer: Respected reviewer thank you for highlighting the required improvement. We have majorly revised our manuscript. Abstract is modified in the revised manuscript according to your suggestions.

Line 30-What is WEP?

Answer: Dear reviewer thank you. Water extractable P. As we used the term available P in the whole manuscript, so we replaced this in the revised manuscript with available P.

Note-Key word are missing.

Answer: Dear reviewer thank you for the comment. Key words are present. In revised manuscript we tried to represent them in better way.

Introduction: The introduction of this manuscript is up to date. However, the authors used much time etc. Instead of etc. put more information that is required to strengthen your manuscript. Kindly do not use etc words.

Answer: Dear reviewer thank you for your comment. We have removed the word etc used in introduction section in the revised manuscript.

Line 37- act as major sink and source (What does it mean?) of key abiotic components such as Carbon

Answer: Key abiotic components are chemical elements such as phosphorus, nitrogen and potassium, these are present in various organic and inorganic forms and convert in other forms such as bioassimilation, when they get absorbed by living microorgansims, MOs are sink here. But when they are released by certain processes or stimuli such as nematodes when prey on bacteria the bacterial phosphorus is released as excretion product from nematodes, this is source of phosphorus.

Line 42- development and growth of plant (Reference?). Agriculture soils are strongly depleted in available P for plants (How??), despite being virtually sufficient in total P 50-65% [3]

Answer: Dear reviewer thank you for highlighting the required improvements and comments.

Yes, we rephrased the sentence to make it comprehensible also provided the reference L38. Agriculture soil are depleted in available P due to; high application of fertilizers, Soil type such as alkaline soil binds P with Ca to form Ca3(PO4)2 and in acidic soil it binds with Iron and Al to from strengite and variscite, , which plant cannot directly asses but in the form of mono and ortho hydrogen phosphate. This means phosphorus is sufficiently present in soil but plants cannot use this complex form of P which is fixed. This sentence is rephrased in revised manuscript.

Line 53- Organic amendments of??

Answer: Respected reviewer thank you. We have completed the sentence in revised manuscript L52.

Line 55-Biomass of??

Dear reviewer thank you for highlighting the required improvement. We have corrected the sentence as” microbial biomass and their diversity” L53 in revised manuscript.

Line 56- other essential ecological functions such as?’

Answer:Dear reviewer thank you we have corrected the sentence as “extracellular enzyme activities e.g. phospahatase enzyme production” L54 in revised manuscript.

Line 61- large amount of P in their biomass (Kindly write the proper sentences. I did not get the point here)

Answer: Dear reviewer thank you for the suggestion. Yes we rephrased the sentence and split in two sentences for better understanding.

Line 64- Depending on the soil and fertilizer input the release of P from microbial biomass-How??

Answer: Yes, Soil MOs adjust according to the soil bioavailability of nutrients. They retain or release P from their biomass not to disturb the CNP ratio of their biomass, so depending upon the fertilizer input more carbon or phosphorus is added to which MOs adapt, more DNA is formed and microorganisms do not retain P in their biomass for longer time.

Question 1: Why did you choose only poultry manure?

Answer: Dear reviewer thank you for the suggestion and comment. We have used other amendments, but we got contradictory results between these amendments based upon their nutrients stoichiometry differences. And the presentation of all amendments data in one manuscript is not possible so here we are presenting the data only from poultry manure.

Question 2: What was the concentration of P in soil before adding PM? And what was the content of P in PM?

Answer: Dear reviewer thank you for the comment. Yes we have analyzed total and available P in soil and PM. We have provided the table 3 in revised manuscript comprising initial analyses of soil and PM regarding P contents.

Question 3: How did you decide the experiments duration for 3-months? Kindly, mention the specific reason.

Answer: Dear reviewer thank you for the valued comments. Our decision to continue the experiment for days was based on the objectives and desired outcomes. Further, we designed a pot experiment on short scale and in small pots, that restrain us to continue the experiment for longer time. Most importantly the effect of microbial inoculation and amendment became evident in the decided duration of the experiment and the effect could be measured well. Due to small capacity of system, it was harvested after 90 days. 

Line 90-91: These microbial growths sequester large amounts of available nutrients such as P, C and N, e.g., nematodes while grazing sequester 50–70% of the prey carbon for biomass production [19]. Kindly rewrite the whole sentence to make it clear for audience.

Answer: Dear reviewer thank you for the valued suggestion. Yes we rephrased the sentence to make it understandable in revised version.

Materials and methods:

Comments-I would like to suggest to the authors kindly make it short and crisp with only required information. Materials and methods are too lengthy as compared to results sections. You can shift unnecessary part of this section to the supplementary sections.

Answer: Dear reviewer thank you for your valuable advice. we have shortened our mat and met section and retained only required information. In revised version now shortened to L111-L 294.

Note-Kindly make the italic all the scientific names.

Answer: Dear reviewer thank you for your suggestion. We have changed all the scientific names to italic font in revised MS.

Line 113- The soil samples were collected from the upper soil layer (depth 113 0-15cm) in triplicates, A substantial P turnover occur in the surface layers of soil and microbial number reduce along soil depth [21]-Why did you write this statement in materials and methods section?

Answer: Dear reviewer thank you. We agree with your comment, so we have removed the sentence from mat and met section and incorporated in introduction section.

Line 119- while the remaining portion of soil was mixed with?

Answer: Dear reviewer thank you for highlighting the required improvement. While improving mat and met section according to the respected reviewer’s suggestions we have rephrased and removed the in comprehensible sentence.

Line 118- Isolation of phosphorus solubilizing bacteria from? And how did you isolate?

Answer: Dear reviewer thank you for the comment. Isolation of PSBs was done from forest rhizosphere soil. We have given a detailed procedure for isolation of PSBs (phosphorus solubilizing bacteria) in the revised manuscript.

Line 120- soil initial pH, water available phosphorus and total phosphorus. All determinations and results were done in triplicate (Kindly make it clear). Line 122- Isolation, screening and identification of PSBs on A (Why this is capital?) synthetic solid medium

Answer: Dear reviewer thank you for the comment. Yes, sentence in the revised manuscript was rephrased. This formatting mistake has been removed in revised manuscript. 

Question 4-How did you control the growth of other microorganisms?

Answer: Dear reviewer thank you for your comment. 

We did not control the growth of soil indigenous microorganisms:

1. We were comparing the organic or plant unavailable P addition as nutrition to MOs (amended) to unamended as close to the natural system.

2. We added treatments to make it close to the natural soil environment because naturally farmers do not control the microbial growth before adding amendment. 

3. On the other hand there were research reports that killing the indigenous flora to control the growth via sterilization of soil can increase the available nutrients from their biomass which can bias the desired results.

Line 165- with each isolate sample by incubating 0.25 mL of water with 1 mL (Space) of pNPP and buffer and Line 166- immediately adding 4 mL (space) of 125 (Space) mM NaOH to stop the reaction. Line 169- 6 efficient strains out of 21 were selected for further analyses (remove extra space). This statement is not supported by your table no-2.

Answer: Respected reviewer thank you for highlighting the required improvements. So we have corrected the formatting mistakes in revised manuscript. 

Dear reviewer thank you for highlighting this inappropriate statement. We have simplified the use of strains from 6 to 4 in revised manuscript in mat and met section, now it is supported by table 2.

Line 177- Phosphorus mineralizing gene expression using q-PCR: In bacteria phoN, phnX gene (full form?) encoding enzymes are responsible for organic P mineralization and gcd (full form?) for inorganic P solubilization i.e.,

Answer: Dear reviewer thank you for the comment. The full forms of primers used are written in revised manuscripts in result section and Table 2.

Note-Figure2-Kindly prepare a new diagram with proper formatting. You must have to keep the size of figures 300 dpi.

Answer: Respected reviewer thank you very much for the valuable suggestion. Yes, after incorporating the reviewer’s all comments we have revised all the figures accordingly and presented in better format. 

Line 227- The synergistic activity by combination of 6, 4, 3 and 2 of 6 or 4?? Selected isolates with high available

Answer: Respected reviewer thank you for highlighting the issue. We have simplified the statement to be comprehensible in mat and met section of revised MS. This sentence was explaining the compatibility test between the strains. Test was performed between 2, 3, 4 isolates.

Figure 1: This is not providing a clear message to the readers; therefore, I would like to ask to make a table for figure 1.

Answer: Dear reviewer thank you for the suggestion. According to the reviewers comments we found it unnecessary and removed it. 

Results:

Note-Kindly, prepare all the figures with proper formation to keep at least 300 dpi size.

Answer: Dear reviewer thank you for your valuable advice. Yes after incorporating the reviewer’s all comments we have revised all the figures accordingly and presented in better format. 

Figure 9-You should replace this figure with other. As I can see clearly shadow instead of bands. I am not able to see the proper bands.

Answer: Dear reviewer thank you for your valuable suggestion. Yes after incorporating the reviewer’s all comments we removed it from the revised MS and mentioned our experiment in results and mat sections.

Note-In each results heading, you should write outcome of the results (kind of 1 line result). For example, do not write Plant dry biomass. You can write treatment of X and Y enhanced dry biomass.

Answer: Dear reviewer thank you for your valuable suggestion. Yes in every section of result where we found significant results, we highlighted them as per your suggestion. 

Note-Conclusion of this manuscript is missing. Kindly write it.

Answer: Conclusion is provided in revised manuscript as per Journal format it is embedded after discussion not as separate section..

Once you address all the comments and queries, after that, I will recommend this manuscript for

further process.

Answer: We addressed all suggestion and revised manuscript and hopeful for better understanding of current version.

---

## [Decision Letter · Decision Letter 1]

18 Mar 2022

Organic amendment plus inoculum drivers: who drives more P nutrition for wheat plant fitness in small duration soil experiment

PONE-D-21-30854R1

Dear Dr. Irshad,

We’re pleased to inform you that your manuscript has been judged scientifically suitable for publication and will be formally accepted for publication once it meets all outstanding technical requirements.

Kind regards,

Academic Editor

PLOS ONE

Additional Editor Comments (optional):

Reviewers' comments:

Reviewer's Responses to Questions

**Comments to the Author**

1. If the authors have adequately addressed your comments raised in a previous round of review and you feel that this manuscript is now acceptable for publication, you may indicate that here to bypass the “Comments to the Author” section, enter your conflict of interest statement in the “Confidential to Editor” section, and submit your "Accept" recommendation.

Reviewer #1: All comments have been addressed

Reviewer #2: All comments have been addressed

2. Is the manuscript technically sound, and do the data support the conclusions?

Reviewer #1: Yes

Reviewer #2: Yes

3. Has the statistical analysis been performed appropriately and rigorously? 

Reviewer #1: Yes

Reviewer #2: Yes

4. Have the authors made all data underlying the findings in their manuscript fully available?

Reviewer #1: Yes

Reviewer #2: Yes

5. Is the manuscript presented in an intelligible fashion and written in standard English?

Reviewer #1: Yes

Reviewer #2: Yes

6. Review Comments to the Author

Reviewer #1: I would like to thank the authors for the improvement of the manuscript after revision. I saw that the authors adressed all comments and suggestions of the reviewer by incorporating them and updating in the text. Thus, I am fully satisfied of authors responses to reviewer's comments. I think that the manuscript meets the journal's requirement now, and could be considered for publication.

Reviewer #2: (No Response)

7. PLOS authors have the option to publish the peer review history of their article (what does this mean?). If published, this will include your full peer review and any attached files.

Reviewer #1: **Yes: **Andriamananjara Andry

Reviewer #2: No

---

## [Editor Report · Acceptance letter]

1 Apr 2022

PONE-D-21-30854R1 

Organic amendment plus inoculum drivers: who drives more P nutrition for wheat plant fitness in small duration soil experiment 

Dear Dr. Irshad:

I'm pleased to inform you that your manuscript has been deemed suitable for publication in PLOS ONE. Congratulations! Your manuscript is now with our production department. 

Kind regards, 

on behalf of

Dr Rashid Nazir 

Academic Editor

PLOS ONE